# The Relationship between Mean Platelet Volume and Neutrophil–Lymphocyte Ratio and Liver Fibrosis in Patients with Chronic Hepatitis B

**DOI:** 10.3390/medicina59071287

**Published:** 2023-07-12

**Authors:** Mehmet Onder Ekmen, Metin Uzman

**Affiliations:** Gastroenterology Department, Ankara Ataturk Sanatoriıum Training and Research Hospital, 06290 Ankara, Turkey; drmetingastro@gmail.com

**Keywords:** chronic hepatitis B, liver fibrosis, MPV (mean platelet volume), NLR (neutrophil to lymphocyte ratio)

## Abstract

*Objective:* The neutrophil–lymphocyte ratio (NLR) can be helpful both in demonstrating acute and chronic liver injury and predicting malignant potential. The mean platelet volume (MPV) is also a marker that can be used as a risk indicator in atherosclerosis-associated diseases, reflecting inflammation. Within the scope of this research, we aimed to elucidate the relationship between the mean platelet volume and neutrophil–lymphocyte ratio in liver fibrosis in chronic hepatitis B patients. *Materials and Methods:* A total of 84 patients who were followed up with the diagnosis of chronic hepatitis B and who underwent liver biopsy were included in this prospective study. Complete blood count parameters (neutrophil, lymphocyte, neutrophil–lymphocyte ratio, hemoglobin, thrombocyte, MCV, and MPV values), demographic data, biochemistry panel (AST, ALT), HBV DNA, and liver biopsy fibrosis scores obtained from hospital database were analyzed. Since the follow-up period of chronic hepatitis B patients is six months, patients were screened in order to include a full 3-year screening pool. *Results:* A total of 84 patients were enrolled within the scope of this study. The chronicity index was ‘0’ in 7.1% (*n* = 6), ‘1’ in 23.8% (*n* = 20), ‘2′ in 56% (*n* = 47), and ‘3’ in 13.1% (*n* = 11)of the patients. According to the chronicity index groups, there was a statistically significant difference between the medians of the hepatitis activity index (HAI) values at the 5% significance level (*p* < 0.001). The correlation analysis revealed a statistically significant relationship between the chronicity index and the mean platelet volume to lymphocyte ratio (MPVL). However, considering the degree of the relationship, it can be said that it was a positive and weak relationship (*p* = 0.046, r = 0.218). *Conclusions:* Regarding the outcomes of this research, a significant relationship was found between the neutrophil–lymphocyte ratio, mean platelet volume, and fibrosis.

## 1. Introduction

Hepatitis B virus (HBV) is still the most common cause of chronic hepatitis, cirrhosis, and hepatocellular cancer (HCC) and is a serious health problem today. It has been reported that carriers of hepatitis B surface antigen (HBsAg) account for approximately 5% of the world’s population. The number of chronic HBV carriers has exceeded approximately 350 million today, and approximately one million patients die from complications caused by HBV infection each year [1].

The disease is progressive in 20–30% of patients with chronic HBV infection and causes cirrhosis with end-stage liver failure [2]. According to current clinical practice, liver biopsy is required to diagnose and treat progressive chronic liver disease. Liver biopsy to evaluate liver damage is still the gold standard today, providing important information about the disease’s histological activity and fibrosis stage. Advanced information about the course of the disease and the treatment results can be derived from the biopsy. However, liver biopsy is an invasive, sometimes complicated, costly, and laborious procedure that requires the opinion of a specialist pathologist [3]. Considering the patient’s reservations about liver biopsy, non-invasive histological indicators are needed to evaluate the disease status and fibrosis in chronic HBV patients.

The idea of utilizing routine biochemical tests in clinical practice may be useful for patients who cannot undergo a liver biopsy. The mean platelet volume (MPV) determines the platelet function and reflects increased activation [4]. Recently, many studies have been published on the mean platelet volume and the clinical significance of various diseases [5]. It has been stated that the neutrophil–lymphocyte ratio can guide the clinician in predicting mortality in acute or chronic viral hepatitis when there is a sign of liver failure and in the determination of HCC recurrence after liver transplantation [6].

The neutrophil–lymphocyte ratio (NLR) and mean platelet volume (MPV) calculations are inexpensive and provide easily accessible markers from routinely processed complete blood counts. It has been stated that the neutrophil–lymphocyte ratio can be a guide in predicting mortality and determining HCC recurrence after liver transplantation when liver failure is present in acute or chronic viral hepatitis. The NLR can be helpful both in demonstrating acute and chronic liver injury and predicting malignant potential [7]. The mean platelet volume (MPV) is also a marker that can be used as a risk indicator in atherosclerosis-associated diseases, reflecting inflammation [8]. Within the scope of this research, we aimed to elucidate the relationship between the mean platelet volume and neutrophil–lymphocyte ratio in liver fibrosis in chronic hepatitis B patients.

## 2. Materials and Method

This prospective research was conducted with 84 chronic hepatitis B patients who applied to Ankara Atatürk Sanatorium Training and Research Hospital Gastroenterology Clinic between 1 January 2020 and 31 December 2022.

All procedures followed were in accordance with the ethical standards of the responsible committee on human experimentation (institutional and national) and with the Helsinki Declaration of 1975, as revised in 2008. The ethics committee approval was granted from our institution with protocol number 2012–KAEK–15/2706, and the informed consent was obtained from all participants.

Patients who were followed up with the diagnosis of chronic hepatitis B and applied to the gastroenterology outpatient clinic and those who underwent liver biopsy were included in this study. Complete blood count parameters (neutrophil, lymphocyte, neutrophil–lymphocyte ratio, hemoglobin, thrombocyte, MCV, and MPV values), demographic data, biochemistry panel (AST, ALT), HBV DNA, and liver biopsy fibrosis scores obtained from hospital database were analyzed. The patients included in this study did not have any additional diseases other than hepatitis B, and they did not have any infectious diseases that would affect liver enzymes and change their blood values.

Since the follow-up period of chronic hepatitis B patients is six months, patients were screened between 1 January 2020 and 31 December 2022 in order to include a full 3-year screening pool. Fibrosis and chronicity indices obtained after liver biopsy were compared with values such as hematological parameters (NLR, PLR, MPV) of the patients, and the effectiveness and reliability of these hematological values in predicting liver fibrosis were examined. The internationally accepted modified Ishak and Knodell scoring systems were used for liver fibrosis scores.

### Statistical Analysis

The data were analyzed via the SPSS (Statistical Package for Social Sciences). While evaluating the study data, descriptive statistical methods’ data (mean, standard deviation) and the measured quantitative data were compared with the Student’s t-test, which showed the normal distribution, and the Mann–Whitney U test, which did not. The Pearson correlation analysis was utilized for parametric values and the Spearman’s correlation analysis for nonparametric values to determine the relationship between the parameters. The results were interpreted at the 95% confidence interval at the *p* < 0.05 significance level.

## 3. Results

A total of 84 patients were enrolled within the scope of this study, with a gender distribution of 77.4% (*n* = 65) male and 22.6% (*n* = 19) female. The patients’ mean age was 67.80 ± 9.5 years (range of 43–92 years), and their laboratory parameters are denoted in Table 1, showing both the mean and median values.

As shown in Table 2, the chronicity index was ‘0’ in 7.1% (*n* = 6), ‘1’ in 23.8% (*n* = 20), ‘2’ in 56% (*n* = 47), and ‘3’ in 13.1% (*n* = 11) of the patients. The chronicity index (CcI) was grouped as 0–1 and 2–3, and intergroup descriptive statistics and comparisons were made for all continuous variables. According to the chronicity index groups, there was a statistically significant difference between the medians of the hepatitis activity index (HAI) values at the 5% significance level (*p* < 0.001) (a three-unit difference between the group medians is considered statistically significant).

As shown in Table 3, the median HAI value of the group with a chronicity index of 2–3 was higher than that of the group with a chronicity index of 0–1. It was observed that there was no statistically significant difference at the 5% significance level in terms of mean and median between the chronicity index groups of other variables.

As shown in Table 4, as a result of the correlation analysis, it was seen that there was a statistically significant relationship at the 5% significance level only between the chronicity index and the mean-platelet-volume–lymphocyte ratio (MPVL) variable. However, considering the degree of the relationship, it can be said that it was a positive and weak relationship (*p* = 0.046, r = 0.218).

## 4. Discussion

It is extremely important to determine the indication for the initiation of treatment in order to prevent the development of complications such as liver fibrosis and HCC in patients with chronic hepatitis B. Regarding the treatment decisions, the first evaluations to be made are the determination of ALT and viral load. After these evaluations, a liver biopsy is performed, or a follow-up with the laboratory values of the patient is conducted in order to start treatment [9,10]. However, cirrhotic patients may sometimes have a low viral load due to advanced cell damage. Therefore, low viral load cannot be used as a marker for liver damage [11]. Hepatitis B infection is still a common problem worldwide and may cause complications such as cirrhosis and hepatocellular cancer in the future. In our country, patients with chronic hepatitis infection must undergo a liver biopsy, which is an invasive method, before starting antiviral therapy. Since liver biopsy can lead to various complications, especially bleeding, the efficacy of noninvasive tests without the risk of complications in the patient in detecting fibrosis in the liver has been investigated recently [4].

In the current research, we have utilized the chronicity index and achieved a statistically significant difference between the medians of the hepatitis activity index (HAI) (*p* < 0.001). The median HAI value of the group with a chronicity index of 2–3 was higher than that of the group with a chronicity index of 0–1.

It has been stated that the neutrophil–lymphocyte ratio and MPV can be used as markers of chronic inflammation in various diseases, such as cardiovascular diseases, malignancies, and liver cirrhosis. Liu et al. affirmed that the NLR values of 2.36 and below were associated with a low mortality risk, and the values of 6.12 and above were associated with a high mortality risk in adult hepatitis B cases with liver failure [12]. Similarly, Chen et al. reported that low NLR values were higher in adult patients with liver failure compared to both the control group and the chronic hepatitis B group [13]. Yılmaz et al. published that the NLR level was significantly lower in adult inactive hepatitis B carriers with a fibrosis degree of two and above [14]. Hu et al. found that the MPV values were higher in patients with chronic severe hepatitis B than in patients with acute hepatitis B, chronic hepatitis B, and in the control groups, and the MPV values in patients with chronic hepatitis B were higher than those in both patients with acute hepatitis B and the control groups [15]. Han et al. indicated higher MPV values in adult hepatitis B patients with liver failure than in patients with chronic hepatitis B, cirrhosis, and in the control groups [16].

Platelets play a primary role in hemostasis. It is known that changes in the platelet parameters occur in chronic liver diseases. It has been reported that with the increase in the MPV, the activation of platelets increases; they aggregate more easily with some cytokines and other mediators they secrete and may cause various diseases [4]. Ekiz et al. reported a statistically significant difference between the patient and control groups regarding the MPV values as 8.49 ± 0.84 in the chronic hepatitis B patients and 7.65 ± 0.42 in the control group [8]. Turhan et al. conducted a study with 260 inactive hepatitis B carriers. They stated that MPV was 8.8 ± 1.2 in the inactive carriers, 8.1 ± 0.9 in the control group, and significantly higher in the inactive HBsAg carrier group [17]. Ceylan et al. divided chronic hepatitis B patients into two groups according to their fibrosis score (group 1: fibrosis score 0–3; group 2: fibrosis score 4–6). In group 1, the MPV was 8.7, and 9.4 in group 2, thus indicating the value of the MPV in providing useful information to predict the degree of liver fibrosis [18]. In the study of Atay, the MPV values for chronic hepatitis B patients were 10.66 ± 1.29 in the patient group, while they were 9.59 ± 0.69 in the control group, with a statistical significance *p* = 0.001. They divided the chronic hepatitis B patients into two groups according to their fibrosis and found that the age of patients with advanced fibrosis was older. The number of female patients was predominant in patients with advanced fibrosis, and it was statistically significant [19]. Our results were similar to those published in the previous literature. As a result of the correlation analysis, the relationship between the chronicity index and the MPV and NLR was not explained in detail, since there was no statistically significant result. However, it was observed that there was a significant relationship at the 5% significance level between the MPVL variable and the chronicity index. Considering the studies published in the literature, no study with a statistically significant result, examining the relationship between the MPVL and the chronicity index, has been found. As a result of the correlation analysis, a statistically significant relationship between the chronicity index and the mean-platelet-volume–lymphocyte ratio (MPVL) was observed.

The neutrophil–lymphocyte ratio (NLR) and mean platelet volume (MPV) calculations are inexpensive and provide easily accessible markers obtained from the complete blood count during daily routine procedures. The NLR is also recognized as a parameter that can guide the prediction of mortality in patients with liver failure in acute or chronic viral hepatitis and the determination of the HCC recurrence after liver transplantation [20].

## 5. Conclusions

In conclusion, it is known that the gold standard in the follow-up and treatment of chronic HBV is liver biopsy. However, performing a biopsy is not always possible, and liver histopathology should be evaluated. It is necessary to use some markers in order to predict liver fibrosis and the disease status of the liver. Regarding the outcomes of this research, a significant relationship was found between the neutrophil–lymphocyte ratio, mean platelet volume, and fibrosis.

## Figures and Tables

**Table 1 medicina-59-01287-t001:** Baseline demographic parameters of the patients.

	N	Mean	Standard Deviation	Median	Minimum	Maximum	IQR
Age	84	67.80	9.50	68.50	43.0	92.0	12
WBC (10 × 3/µL)	84	7.77	2.89	7.40	3.7	15.2	4.1
Neutrophil (%)	84	4.86	1.95	4.74	1.80	10.86	2.74
Lymphocyte (%)	84	2.22	0.86	2.12	0.64	4.88	1.3
Hemoglobin (gr/dL)	84	10.90	1.61	10.70	6.4	14.2	2.6
Platelets (mm^3^)	84	231.89	82.58	240.40	82.4	420.0	100.8
MPV (fL)	84	9.43	1.44	9.40	6.4	13.0	1.8
Monocytes (%)	84	0.162	0.201	0.100	0.002	1.000	0.18
Albumin (gr/dL)	84	31.91	6.47	32.00	16.6	46.0	10
Protein (gr/dL)	84	62.88	9.39	64.60	28.00	78.00	12.35
AST (IU/L)	84	54.29	28.45	46.00	11.0	136.0	36
ALT (IU/L)	84	57.73	30.38	51.00	12.0	146.0	36
LDH (IU/L)	84	207.15	93.51	187.00	112.0	782.0	100
NLR	84	2.31	0.93	2.05	1.31	5.75	0.81
MCV (fL)	84	79.85	11.58	80.40	54.4	106.8	16.2
HbeAg (IU)	84	1.24	3.02	0.367	0.120	14.364	0.37
HBVDNA (IU/mL)	84	166.08	354.13	45.32000	10.00	2440.00	129.81
HAI	84	6.869	2.8232	6.000	2.0	16.0	3
PLR	84	121.31	71.13	103.73	31.43	503.13	80.53
MPVP	84	0.05	0.02	0.04	0.02	0.12	0.03
MPVL	84	5.03	2.48	4.40	1.90	17.19	3.37

**Table 2 medicina-59-01287-t002:** Comparison of the data of patients grouped by chronicity index.

	Chronicity Index	N	Mean	Standard Deviation	Median	Minimum	Maximum	Interquartile Range	*p*-Value
Age	0–1	26	69.69	10.44	68.00	51.00	92.00	13.30	0.223
	2–3	58	66.95	9.01	68.50	43.00	86.00	13.50	
WBC	0–1	26	7.96	2.66	7.25	4.60	13.90	3.90	0.536
	2–3	58	7.68	3.01	7.40	3.70	15.20	4.20	
Neutrophils	0–1	26	4.85	1.59	4.58	2.20	8.80	2.74	0.692
	2–3	58	4.87	2.10	4.74	1.80	10.86	2.81	
Lymphocyte	0–1	26	2.40	0.76	2.15	1.24	4.10	1.13	0.141
	2–3	58	2.13	0.89	2.11	0.64	4.88	1.36	
Hemoglobin	0–1	26	11.12	1.56	10.90	8.20	13.60	2.50	0.410
	2–3	58	10.80	1.64	10.60	6.40	14.20	2.60	
Platelets	0–1	26	257.37	89.96	249.30	88.00	420.00	136.30	0.058
	2–3	58	220.47	77.16	225.65	82.40	412.20	102.00	
MPV	0–1	26	9.30	1.32	9.20	6.40	12.60	1.50	0.568
	2–3	58	9.49	1.50	9.55	6.40	13.00	2.40	
Monocytes	0–1	26	0.19	0.22	0.14	0.00	1.00	0.17	0.186
	2–3	58	0.15	0.19	0.06	0.00	0.88	0.19	
Albumin	0–1	26	32.51	6.19	34.00	18.00	40.00	7.60	0.427
	2–3	58	31.64	6.62	32.00	16.60	46.00	10.10	
Tprotein	0–1	26	63.73	8.64	65.30	40.20	76.40	9.53	0.517
	2–3	58	62.50	9.75	64.40	28.00	78.00	13.50	
AST	0–1	26	55.54	26.44	51.00	16.00	114.00	41.00	0.574
	2–3	58	53.72	29.51	46.00	11.00	136.00	36.50	
ALT	0–1	26	58.42	27.44	53.00	18.00	120.00	43.50	0.588
	2–3	58	57.41	31.84	50.00	12.00	146.00	36.50	
LDH	0–1	26	200.73	57.39	188.00	130.00	340.00	96.50	0.674
	2–3	58	210.03	106.12	177.00	112.00	782.00	111.80	
NLR	0–1	26	2.05	0.42	2.03	1.35	3.19	0.49	0.34
	2–3	58	2.43	1.07	2.06	1.31	5.75	0.98	
MCV	0–1	26	83.02	11.92	82.50	64.40	106.80	18.70	0.093
	2–3	58	78.42	11.23	79.10	54.40	104.00	16.60	
HbeAg	0–1	26	1.50	3.21	0.36	0.15	12.36	0.37	0.632
	2–3	58	1.12	2.95	0.39	0.12	14.36	0.36	
HBVDNA	0–1	26	254.80	515.13	45.30	10.00	2440.00	198.85	0.720
	2–3	58	126.31	247.03	45.44	10.21	1620.00	128.08	
HAI	0–1	26	4.12	0.82	4.00	2.00	5.00	1.00	0.000
	2–3	58	8.10	2.51	7.00	6.00	16.00	4.00	
PLR	0–1	26	119.51	60.05	111.42	31.43	245.97	87.17	0.877
	2–3	58	122.11	76.05	100.01	41.24	503.13	80.14	
MPVP	0–1	26	0.04	0.02	0.04	0.02	0.11	0.027	0.105
	2–3	58	0.05	0.02	0.05	0.02	0.12	0.026	
MPVL	0–1	26	4.20	1.33	3.80	2.61	7.21	2.36	0.098
	2–3	58	5.40	2.78	4.78	1.90	17.19	3.95	

**Table 3 medicina-59-01287-t003:** Comparison of the data of patients divided into two groups according to HAI score.

	HAI	N	Mean	Standard Deviation	Median	Minimum	Maximum	Interquartile Range	*p*-Value
Age	HAI < 6	26	69.69	10.44	68.00	51.00	92.00	13.30	0.223
	HAI ≥ 6	58	66.95	9.01	68.50	43.00	86.00	13.50	
WBC	HAI < 6	26	7.96	2.66	7.25	4.60	13.90	3.90	0.536
	HAI ≥ 6	58	7.68	3.01	7.40	3.70	15.20	4.20	
Neutrophils	HAI < 6	26	4.85	1.59	4.58	2.20	8.80	2.74	0.692
	HAI ≥ 6	58	4.87	2.10	4.74	1.80	10.86	2.81	
Lymphocyte	HAI < 6	26	2.40	0.76	2.15	1.24	4.10	1.13	0.141
	HAI ≥ 6	58	2.13	0.89	2.11	0.64	4.88	1.36	
Hemoglobin	HAI < 6	26	11.12	1.56	10.90	8.20	13.60	2.50	0.410
	HAI ≥ 6	58	10.80	1.64	10.60	6.40	14.20	2.60	
Platelets	HAI < 6	26	257.37	89.96	249.30	88.00	420.00	136.30	0.058
	HAI ≥ 6	58	220.47	77.16	225.65	82.40	412.20	102.00	
MPV	HAI < 6	26	9.30	1.32	9.20	6.40	12.60	1.50	0.568
	HAI ≥ 6	58	9.49	1.50	9.55	6.40	13.00	2.40	
Monocytes	HAI < 6	26	0.19	0.22	0.14	0.00	1.00	0.17	0.186
	HAI ≥ 6	58	0.15	0.19	0.06	0.00	0.88	0.19	
Albumin	HAI < 6	26	32.51	6.19	34.00	18.00	40.00	7.60	0.427
	HAI ≥ 6	58	31.64	6.62	32.00	16.60	46.00	10.10	
Tprotein	HAI < 6	26	63.73	8.64	65.30	40.20	76.40	9.53	0.517
	HAI ≥ 6	58	62.50	9.75	64.40	28.00	78.00	13.50	
AST	HAI < 6	26	55.54	26.44	51.00	16.00	114.00	41.00	0.574
	HAI ≥ 6	58	53.72	29.51	46.00	11.00	136.00	36.50	
ALT	HAI < 6	26	58.42	27.44	53.00	18.00	120.00	43.50	0.588
	HAI ≥ 6	58	57.41	31.84	50.00	12.00	146.00	36.50	
LDH	HAI < 6	26	200.73	57.39	188.00	130.00	340.00	96.50	0.674
	HAI ≥ 6	58	210.03	106.12	177.00	112.00	782.00	111.80	
NLR	HAI < 6	26	2.05	0.42	2.03	1.35	3.19	0.49	0.34
	HAI ≥ 6	58	2.43	1.07	2.06	1.31	5.75	0.98	
MCV	HAI < 6	26	83.02	11.92	82.50	64.40	106.80	18.70	0.093
	HAI ≥ 6	58	78.42	11.23	79.10	54.40	104.00	16.60	
HbeAg	HAI < 6	26	1.50	3.21	0.36	0.15	12.36	0.37	0.632
	HAI ≥ 6	58	1.12	2.95	0.39	0.12	14.36	0.36	
HBVDNA	HAI < 6	26	254.80	515.13	45.30	10.00	2440.00	198.85	0.720
	HAI ≥ 6	58	126.31	247.03	45.44	10.210	1620.00	128.08	
Chronicity Index	HAI < 6	26			1	0	1	0.30	0.000
	HAI ≥ 6	58			2	2	3	0.00	
PLR	HAI < 6	26	119.51	60.05	111.42	31.43	245.97	87.17	0.877
	HAI ≥ 6	58	122.11	76.05	100.01	41.24	503.13	80.14	
MPVP	HAI < 6	26	0.04	0.02	0.04	0.02	0.11	0.027	0.105
	HAI ≥ 6	58	0.05	0.02	0.05	0.02	0.12	0.026	
MPVL	HAI < 6	26	4.20	1.33	3.80	2.61	7.21	2.36	0.098
	HAI ≥ 6	58	5.40	2.78	4.78	1.90	17.19	3.95	

**Table 4 medicina-59-01287-t004:** Correlation of HAI and chronicity index with other variable levels.

		NLR	PLR	MPV	MPVP	MPVL	HbeAg	HBVDNA
HAI	r	0.212	0.045	0.096	0.152	0.215	0.128	−0.121
	*p*	0.052	0.681	0.385	0.167	0.050	0.245	0.271
	*n*	84.000	84.000	84	84	84	84	84
Chronicity Index	r	0.121	0.076	0.074	0.101	0.218	0.090	−0.028
	*p*	0.272	0.493	0.503	0.360	0.046	0.413	0.802
	*n*	84	84	84	84	84	84	84

## Data Availability

The data used in this study are available upon reasonable request.

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
