# Peer review of "The Relationship between Mean Platelet Volume and Neutrophil–Lymphocyte Ratio and Liver Fibrosis in Patients with Chronic Hepatitis B"

_medicina, 2023, doi:10.3390/medicina59071287_

Round 1

Reviewer 1 Report

Dear colleagues,

Generally your paper is original and interesting but needs some additional checking. As an example: "Gender distribution was as follows: 7.4% (n=65) were male, and 22.6% (n=19) were female." Did you mean 77.4 instead of 7.4? I think so. Please check text again.

Next remark. Isn't clear about comorbidity of your patients group: are they HBV-positive alone or some of them have hepatitis B+C or other types of liver related comorbidities? No data about laboratory examination for other infectious diseases which can affect liver and change blood parameters.

Please, specify measurement units in Table 1. Did you use genomes (copies) per ml for HBV DNA or other units? How you measured HBeAg? In IU or other scale? It will be useful for readers.

Nevertheless your paper is interesting due to topic importance because we all need precise, low invasive and informative marker for liver fibrosis. Cheap and easy to perform assays like blood parameters count could be perspective for mass-screening amongst patients with hepatitis B.  

Language is clear and understandable, no comments

Author Response

Response to reviewer 1 comments

Dear reviewer

Point 1: Generally your paper is original and interesting but needs some additional checking. As an example: "Gender distribution was as follows: 7.4% (n=65) were male, and 22.6% (n=19) were female." Did you mean 77.4 instead of 7.4? I think so. Please check text again.

Reponse 1:Yes, you are right.I calculated the the distribution again and corrected the result on page 7

Point 2:Next remark. Isn't clear about comorbidity of your patients group: are they HBV-positive alone or some of them have hepatitis B+C or other types of liver related comorbidities? No data about laboratory examination for other infectious diseases which can affect liver and change blood parameters.

Response 2:The patients included in the study did not have any additional diseases other than hepatitis B, and they did not have any infectious diseases that would affect liver enzymes and change their blood values

Point 3:Please, specify measurement units in Table 1. Did you use genomes (copies) per ml for HBV DNA or other units? How you measured HBeAg? In IU or other scale? It will be useful for readers.

Response 3:Necessary corrections were made in the tables.

Point 4:Nevertheless your paper is interesting due to topic importance because we all need precise, low invasive and informative marker for liver fibrosis. Cheap and easy to perform assays like blood parameters count could be perspective for mass-screening amongst patients with hepatitis B.

Response 4: Thank you dear reviwer, I agree with you. Hematological parameters with noninvasive, inexpensive and informative features are an important consideration in predicting liver fibrosis.

Thank you 

Best regards

Reviewer 2 Report

MPV and NLR were major markers the authors focused on in patients with CHB (n=84). The idea of hematological markers in the course of liver fibrosis still remains a significant phenomenon. MPV and NLR are commonly known in this field. Nevertheless, the direct relationship between MPV and lymphocytes seems to be uninvestigated. I wonder what was the origin of the idea to observe dependences between these exact markers? Secondly, how to explain the presence of a significant relationship between them in the context of liver fibrosis? I would like the authors to expand this topic and to refer to already published data with MPVL (if they exist).

In my opinion the quality of English is acceptable

Author Response

Response to Reviewr 2 Comments

Dear reviewer

Point 1: MPV and NLR were major markers the authors focused on in patients with CHB (n=84). The idea of hematological markers in the course of liver fibrosis still remains a significant phenomenon. MPV and NLR are commonly known in this field. Nevertheless, the direct relationship between MPV and lymphocytes seems to be uninvestigated. I wonder what was the origin of the idea to observe dependences between these exact markers?

Response 1: As a result of the correlation analysis, the relationship between the chronicity index and the MPV NLR was not explained in detail since there was no statistically significant result, but it was observed that there was a significant relationship at the 5% significance level between the MPVL variable and the chronicity index

Point 2:how to explain the presence of a significant relationship between them in the context of liver fibrosis? I would like the authors to expand this topic and to refer to already published data with MPVL (if they exist).

Response 2:Considering the studies published in the literature, when the relationship between MPVL and the chronicity index is examined, no study with a statistically significant result was found

Thank you 

Best regards
